# Palliative Care Knowledge and Attitudes towards End-of-Life Care among Undergraduate Nursing Students at Al-Quds University: Implications for Palestinian Education

**DOI:** 10.3390/ijerph19159563

**Published:** 2022-08-03

**Authors:** Abdallah Ahmad Alwawi, Hammoda Abu-Odah, Jonathan Bayuo

**Affiliations:** 1Department of Nursing, College of Health Professions, Al-Quds University, Abu Dies, Jerusalem P.O. Box 51000, Palestine; aalwawi@staff.alquds.edu; 2Anesthesia and Resuscitation Technology Department, College of Health Professions, Al-Quds University, Abu Dies, Jerusalem P.O. Box 51000, Palestine; 3School of Nursing, The Hong Kong Polytechnic University, Hung Hom, Kowloon, Hong Kong 999077, China; jonathan.bayuo@connect.polyu.hk; 4Centre for Advancing Patient Health Outcomes: A JBI Affiliated Group, School of Nursing, The Hong Kong Polytechnic University, Kowloon, Hong Kong 999077, China; 5Nursing and Health Sciences Department, University College of Applied Sciences (UCAS), Gaza 1415, Palestine

**Keywords:** nursing students, knowledge, palliative care, attitudes, Palestinian, end-of-life care, cross-sectional study

## Abstract

(1) Background: Nursing students should be well-prepared and educated in order to provide high-quality palliative care (PC) to patients with chronic diseases, which will have an impact on the quality of care for dying patients in the future; this study aimed to investigate the level of knowledge about PC and attitudes towards the care of dying patients among Palestinian nursing undergraduate students; (2) Methods: The study was a descriptive quantitative cross-sectional study design of 410 nursing students who participated and completed the questionnaire. The PC Quiz for Nurses (PCQN) and the Short Version of the Frommelt Attitudes Towards Care of the Dying (FATCOD) Form B Scales were used to assess students’ PC knowledge and attitudes toward PC and caring for dying patients. Generalized linear regression analysis was applied to identify the influencing variables on knowledge and attitudes; (3) Results: The overall knowledge mean score was 7.42 ± 2.93, ranging from 0 to 20, which indicates that nursing students lacked knowledge about PC; however, they have a positive attitude (25.94 ± 4.72; ranging from 9 to 45) toward care for a dying person. Receiving lectures or training about PC, caring for relatives in their last days of life, gender, and year of study were independently associated with students’ knowledge and attitudes about PC and care of dying patients; and (4) Conclusion: This study generated evidence showing insufficient knowledge about PC among nursing students at the Al-Quds University, Palestine, albeit a favorable attitude towards EoLC was shown. Integrating PC education into the nursing curriculum at Palestinian Universities need to increase their awareness of PC.

## 1. Introduction

Palliative care (PC) is considered one of the most holistic approaches required to provide comprehensive healthcare and address the needs of individuals of all ages with health-related suffering [1] and make the individuals’ life more meaningful and peaceful. The demand for PC services has surged due to the global population’s significant transitional demographic and disease changes [2] and the impact of the evolving pandemic on national healthcare system services [3]; thus, the World Health Organisation (WHO) categorizes PC as a fundamental human right that should be provided for all human beings regardless of their disease type, income and age, and based on need [4].

Although the WHO and international organizations have applied efforts to provide PC within the countries’ healthcare systems [5,6], there is still a significant unmet need for PC services, particularly in low-and-middle-income countries (LMICs) [5,6]. The recent global atlas of PC 2020 reported that 76% of adults living in LMICs require PC [7]. The need for PC at the end of life (EOL) is expected to double by 2060 [6], with the greatest rise in LMICs. Thus, LMICs should be prepared and ready for the future needs of PC in developing PC policies, ensuring the availability of medications, and integrating PC within undergraduate universities’ health curricula [5].

Undergraduate nursing students who will be future nurses should be well-prepared and well-educated to deliver high-quality PC to patients with chronic illnesses; their knowledge about PC and attitudes towards care of dying will affect the future quality of care for dying patients, and indeed they play a pivotal role in providing PC to terminally ill patients because they spend most of their time at patients’ bedsides [8]. Nursing education is crucial in preparing nurses to provide high-quality care and lifelong PC [9]. The lack of nurses’ knowledge is one of the main challenges to delivering PC services [10]. Only 57.9% of undergraduate nursing students received sessions in PC during their education, and 42.1% have a misunderstanding about PC and pain management [11]. Thus, integrating PC into undergraduate universities’ health curricula is pivotal for mitigating challenges and successfully providing PC services.

Undergraduate student nurses should receive comprehensive PC courses and training to ensure that they have sufficient knowledge and skills necessary to deliver high-quality care to seriously ill and dying patients [12]. Education of nursing students on PC should start from the early stages of education to help and improve their capability to provide optimal care and to perform major PC tasks, including counseling, communication, coordination and information sharing, provision of symptoms relief and psychological support to the patients and their family members [13]. Improving students’ knowledge about PC and changing attitudes through PC education is essential. The attitudes towards care of dying people are psychological behaviors learned as a part of an individual’s cultural experiences during their lifetime; such positive attitudes are very crucial for nurses and can be influenced by professional education and clinical work experiences.

Several studies have been conducted in recent times to assess the undergraduate nursing students’ knowledge about PC, which reported inadequate knowledge about them [13,14,15]. For instance, Aboshaiqah [13] assessed the Saudi Arabia nursing students’ knowledge level of PC and found insufficient knowledge among nurses. Although PC was introduced in Saudi Arabia 30 years ago [12], the efforts towards developing nursing curricula towards PC still weak and do not reflect the importance of PC. Dimoula et al. [14] evaluated the Greek undergraduate nursing students’ knowledge and attitudes about PC and end-of-life care (EoLC) and found low levels of knowledge in symptom management and psychosocial/spiritual care. The Greek students also reported less positive attitudes in relation to student comfort with care of dying patients and their imminent death. Jiang et al. [15] investigated the knowledge and attitudes of Chinese undergraduate nursing students’ towards PC. The findings revealed insufficient knowledge of PC among Chinese students. Despite the findings of the previous studies, each country has a unique context that affects the development of PC, such as the interest of policymakers in PC, development of educational systems, culture, and religion.

In Palestine, little effort has been made regarding PC education, and most of efforts have focused on nurses and physicians working in hospitals [16]. Only one study [17] conducted in Palestine in 2015 examined 141 nursing students’ attitudes toward EoLC; the results reported low levels of attitude towards caring for dying patients and their families. Considering the significant shortage of PC nurses in Palestine and the importance of PC, there is a necessity to educate and train Palestinian nursing students to enhance nursing practices, provide quality services to patients, and overcome the shortage of nursing staff. Some PC courses were provided to students’ nurses, but their content about PC and EoLC were inconsistent and limited. Providing a compressive PC courses is urgently in needed and to be officially integrated with Palestinian nursing universities curricula. Thus, this study was conducted to investigate the level of knowledge about PC and their attitudes towards the care of dying patients and their influencing factors among Al-Quds university nursing undergraduate students; this study will provide a pool of evidence to help identify knowledge and training gaps and areas for improvement in teaching nursing students at Al-Quds university about PC and their attitudes toward the care of dying patients.

## 2. Materials and Methods

### 2.1. Study Design

A descriptive quantitative cross-sectional study design was employed in this study; this study has been reported following the Strengthening the Reporting of Observational Studies in Epidemiology (STROBE) guidelines [18].

### 2.2. Sampling and Participants

A convenience sampling method was used to select the eligible students. Participants were eligible if they (i) were enrolled at Al-Quds University; (ii) were at the end of the 2nd, 3rd, or 4th academic; and (iii) were willing to participate in the study and sign the consent form. Participants were excluded if they (i) were at the first-year of academic, (ii) were not studied at A-Quds University, and (iii) were registered at nursing upgrading program. The first-year nursing students were excluded because they were not involved in any clinical exposure to palliative and end-of-life care, as the nursing programs for the first years were focused only on the taught general nursing courses.

In total, there were 1500 nursing students registered at Al-Quds University. Utilizing the sample size calculation formula described by Thompson [19], the required sample size was 306 participants. In this study, 410 nursing students participated and completed the questionnaire.

### 2.3. Procedures

From February to April 2022, nursing students were recruited from the School of Nursing at Al-Quds University. After obtaining the required approvals from the Nursing Department and the Ethics Committee, the study questionnaire was sent to the students’ official email through the University’s Information Technology Department, and the students were informed that participation is completely voluntary and anonymous, and that non-participation will not lead to any penalty at all. All information security measures have been taken into account to preserve the privacy of the participants.

The PC Quiz for Nurses (PCQN) used to assess student knowledge, the Short Version of the Frommelt Attitudes Towards Care of the Dying (FATCOD) Form B that used for assessing students’ attitudes and the demographic part were sent to 1500 nursing students registered at Al-Quds University to be filled out. The Google platform was used to create and send a link for the questionnaire to the 2nd, 3rd, and 4th academics. Students were asked to fill out a questionnaire. 410 nursing students participated and completed the questionnaire. The completed questionnaire was received from the students within 2–20 days from the date it was sent to them.

### 2.4. Instruments

Two instruments were utilized to collect the data from participants. Permission to use the instruments was obtained prior to data collection from the authors. The instruments were distributed to the participants in English, as the student’s language instruction was in English; the instruments were previously used in an Arab context and fit with the culture.

The PCQN English version was used to assess students’ PC knowledge [20]; this instrument tool that comprises 20 items was designed to assess learning needs in undergraduate nursing programs since gaps in knowledge can indicate where additional material should be included, and misconceptions indicate where reinforcement or repetition is required. The overall score of PCQN ranges from 0 “*lowest level of knowledge*” to 20 “*highest level of knowledge*”. The final answers are coded 1 = “*correct”*, 0 = “*incorrect”*, and “*I do not know”*. The internal consistency of the PCQN was 0.68, indicating an acceptable level of reliability.

The FATCOD- Form B Scale English version was used to assess the students’ attitudes toward caring for dying patients [21]. FATCOD-B consists of 9 items on a 5-point Likert scale Ranging from 1 to 5, where 1 = “*strongly disagree*”, 2 = “*disagree*”, 3 = “*uncertain*”, 4 = “*agree*”, 5 = “*strongly agree*”. The 9 items were with content on negative attitudes. All items were revised to positive content. The mean scores were then calculated and converted to a percentage, and the higher rate reflects a more positive attitude toward the care of dying. The internal consistency of the FATCOD-B in this study was 0.70, indicating an acceptable level of reliability.

The demographic characteristics of the nursing students were also collected, including age, gender, GPA, academic year, received education or training in PC, witnessed dying patients, caring for a family member or relative in their last days of life, and participating in the preparation of dead patients.

### 2.5. Data Analysis

The collected data were analyzed using the Statistical Package for Social Sciences (SPSS) Version 25.0. Data entry was performed and double-checked for outliers or errors. Descriptive statistics, including frequency, percentages, mean score, and Standard Deviation (SD), were used to analyze students’ demographic variables and responses. Cronbach alpha was used to measure the internal consistency of the PCQN and FATCOD-B instruments, and alpha ranging between 0.6 and 0.7 indicating an acceptable level of reliability, and 0.8 or greater a very good level. Univariate analyses [An independent sample *t*-test and analysis of variance (ANOVA)] were used as appropriate to examine the differences among knowledge, attitude, and demographic variables. All variables with a *p*-value less than 0.25 in univariate analysis were selected for generalized linear regression to identify the influencing variables on knowledge and attitudes. *p* < 0.05 was considered statistically significant.

## 3. Results

### 3.1. Characteristics of the Nursing Students

Of the 450 questionnaires distributed to students, 418 were collected. Out of 418 collected questionnaires, eight questionnaires were excluded because of significant missing data. A total of 410 participants were included in the final analysis, with a response rate of 91.1%. Half of the student nurses (n = 210, 51.2%) were females. Two-thirds of students (n = 273, 66.6%) were aged more than 20 years old, with a mean age of 21.36 ± 2.15. More than half of the students (n = 219, 53.4%) were in the 4th year of their study.

About 249 student nurses (n = 60.7%) witnessed dying patients during their academic training in the hospitals. Half of the students (n = 205) cared for a family member on their last day of life. More than half of the students (n = 233) reported that they were given lectures or training about PC in their undergraduate program, and 188 (45.9%) of students participated in the preparation of dead patients (Table 1).

### 3.2. Knowledge of Nursing Students regarding PC

Overall knowledge mean score was 7.42 ± 2.93, ranging from 0 to 20, and the correct average rate was 37.1%, indicating that nursing students lacked knowledge about PC (Table 2). The students also reported insufficient PC knowledge in all PCQN categories, mainly in psychological and spiritual care and philosophy and principles of PC aspects (correct rate 24.5% and 30.3%, respectively). The responses of participants’ to PCQN items are presented in Table 2.

### 3.3. Attitudes of Healthcare Professionals toward the Care of Dying

FATCOD-B Questionnaire’s total score was 25.94 ± 4.72, ranging from 9 to 45. The average indicated that 57.6% of students had positive attitudes toward care for dying. For FATCOD-B items, results revealed that nursing students have a positive attitude toward care for a dying person with 3.59 ± 1.06. The details responses of students about caring for dying items are presented in Table 3.

### 3.4. Factors Associated with Nursing Students’ Knowledge about Palliative Care and Attitudes toward Care of Dying

The generalized linear regressions analyses were utilized to predict factors and influence nursing students’ knowledge and attitudes (Table 4). Variables with *p* values less than 0.25 in univariate analysis were considered in the regression analysis. The findings underscored that receiving lectures or training about PC in our study (β = 1.002; *p*-value = 0.001) and students’ GPA (β = 0.031; *p*-value = 0.016) were independently associated with nursing student’s knowledge towards PC. Caring of relative in their last days of life (β = 0.962; *p*-value = 0.049), gender (β = 2.071; *p*-value = 0.000), year of study (β = 1.487; *p*-value = 0.017) and receiving lectures or training about PC in your study (β = 1.410; *p*-value = 0.004) were independently associated with attitudes about EoLC (Table 4).

## 4. Discussion

This is the first study conducted in Palestine to assess undergraduate nursing students’ level of knowledge about PC and their attitudes toward the care of dying patients. The findings suggest insufficient knowledge about PC among Palestinian nursing students; however, they had favorable attitudes towards caring for dying patients. Receiving lectures or training about PC, students’ grade point average (GPA), caring for relatives in their last days of life, gender, and year of study were independently associated with students’ knowledge and attitudes about PC and EoLC.

Palestinian nursing students reported insufficient knowledge about PC; this finding matches with what has been reported in recent studies that reported low knowledge among nursing students [13,14,22]; however, the mean PC score is slightly lower than the studies as mentioned above that ranged their knowledge between 5.23–10.41, compared with 2.88 in this study. The low PC knowledge among Palestinian students could justify the unavailability/lack of PC education and training in students’ curricula. The previous study has pointed out the increasing need for PC education to enhance the students’ knowledge and awareness of PC [23]. Thus, there is an urgent need for accreditation bodies to integrate PC into the Palestinian university educational curricula, such as the Ministry of the Higher Education Accreditation Commission and the nursing council.

A favorable attitude towards the care of dying patients was observed in the current study. Attitude has a pivotal influence on the quality of PC services provided [16]; this result is in line with previous studies conducted In Muslim countries, including Palestine [17] and Turkey [24]; however, the attitude scores are slightly lower than those reported in China Mainland [22], the USA [25] and Sweden [26]; however, it contradicts another study conducted in China which reported negative attitudes among nursing students [15]. The differences across studies might be attributed to educational, religious and cultural differences across nations [27]. The favorable attitude among Palestinian nursing students justified the Islamic religious beliefs that acknowledge the inevitability of death and accept death as a normal process; however, the attitudes of Palestinian students could be strengthening through enhancing their knowledge about PC. The variability between countries suggests how culture can influence the formation of attitudes regarding death and dying. The slightly high score of attitudes in the USA, Sweden, and China might be ascribed to the long development history of PC research and policies. For instance, in Sweden, most universities provide PC education for nursing students [26]. In Palestine, PC has not yet been integrated into nursing curricula. In 2015, PC was only incorporated into the curriculum of the Faculty of Medicine at the Islamic University as an intensive short subject for five days, including hospital training [28].

Having cared for relatives in their last days of life can significantly predict nursing students’ attitudes towards the care of dying patients, as reported in this study; this result is congruent with previous studies [26,29]. Furthermore, receiving lectures or training about PC was independently associated with nursing students’ knowledge and attitudes towards PC and care for dying. Such factors-experiences can strongly affect students’ knowledge and attitudes toward PC and EoLC. We found that male nursing students reported positive attitudes towards the care of dying patients; this result is matched to a previous study [14]. It is difficult to explain why males had more positive attitudes than female students, which needs further investigation. Furthermore, there was no difference between male and female nursing students about their knowledge of PC; this result contradicted Chinese [15] and Greece [14] studies that reported high knowledge among female students. The absence of relation among gender in this study may be justified by the unexposed of students to any PC education and training program.

This study contributes to the development of PC nursing practice in Palestine. Adopting and integrating an appropriate nursing PC educational programs’ within Al-Quds University nursing curricula is essential to enhance the students’ knowledge about PC. The educational PC programs should be tailored to the Islamic cultural and religion backgrounds. Improving nursing students’ knowledge and attitudes will positively contribute to the healthcare services provided to EOL patients. More clinical practice focus should be paid on strengthening nursing students’ ability to psychologically and emotionally deal with the challenges in the process of patient’s dying.

There are several limitations of this study. Adopting cross-sectional research makes it difficult to examine the causal relationship among variables. Furthermore, adopting a convenience sampling approach makes generalizing the findings difficult. Using the English version of the questionnaire could hinder student nurses’ comprehension and understanding of some items. Allowing the participants to fill the questionnaire online may affect the accuracy of the results as some of the students may be looked up on the answer to the questionnaire items on the internet; however, the large sample size of this study enhances the generalization of these findings. As aforementioned, this is the first study assessing nursing students’ knowledge and attitudes, which provides information on the gap and the areas that need strengthening and improvement.

## 5. Conclusions

This study generated evidence showing insufficient knowledge about PC among Palestinian nursing students at Al-Quds University, albeit favorable attitude towards EoLC. Insufficient knowledge could be due to the inadequate PC education in the nursing program. Students’ attitudes were positively influenced by the care of a family member or relative in their last days of life. Enhancing the students’ knowledge could, therefore positively improve the students’ attitudes towards death and care of dying people; this study recommends integrating PC education into the nursing curriculum at Palestinian Universities. Future studies can design PC courses to improve students’ knowledge and competence in delivering PC at universities.

## Figures and Tables

**Table 1 ijerph-19-09563-t001:** Socio-demographic characteristics of nursing students (n = 410).

Variables	Categories	N	%
Gender	Male	200	48.8
	Female	210	51.2
Age group	≤20 years old	137	33.4
	>20 years old	273	66.6
Year of the study	2nd year	115	28.0
	3rd year	76	18.6
	4th year	219	53.4
Cumulative grade point average	60–69.9%	11	2.7
	70–79.9%	148	36.1
	80–89.9%	242	59.0
	≥90%	9	2.2
Have you been given lectures or training about palliative care in your study?	Yes	233	56.8
	No	177	43.2
Have seen or witnessed a dying patient?	Yes	249	60.7
	No	161	39.3
Have you cared of a family member or relative in their last days of life?	Yes	205	50.0
	No	205	50.0
Have you prepared or participated in the preparation of dead patients?	Yes	188	45.9
	No	222	54.1

**Table 2 ijerph-19-09563-t002:** Distribution of nursing students’ knowledge about palliative care (n = 410).

No.	Item	Correct Answers		Incorrect Answers/Did Not Know	
		N	%	n	%
Q1	Palliative care is appropriate only in situations where there is evidence of a downhill trajectory or deterioration. (F)	181	44.1	229	55.9
Q2	Morphine is the standard used to compare the analgesic effect of other opioids. (T)	208	50.7	202	49.3
Q3	The extent of the disease determines the method of pain treatment. (F)	61	14.9	349	85.1
Q4	Adjuvant therapies are important in managing pain. (T)	325	79.3	85	20.7
Q5	It is crucial for family members to remain at the bedside until death occurs. (F)	113	27.6	297	72.4
Q6	During the last days of life, the drowsiness associated with electrolyte imbalance may decrease the need for sedation. (T)	111	27.1	299	72.9
Q7	Drug addiction is a major problem when morphine is used on a long-term basis for the management of pain. (T)	67	16.3	343	83.7
Q8	Individuals who are taking opioids should also follow a bowel regime (laxative treatment). (T)	188	45.9	222	54.1
Q9	The provision of palliative care requires emotional detachment. (F)	152	37.1	258	62.9
Q10	During the terminal stages of an illness, drugs that can cause respiratory depression are appropriate for the treatment for severe dyspnea. (T)	151	36.8	259	63.2
Q11	Men generally reconcile their grief more quickly than women. (F)	109	26.6	301	73.4
Q12	The philosophy of palliative care is compatible with that of aggressive treatment. (T)	162	39.5	248	60.5
Q13	The use of placebos is appropriate in the treatment of some types of pain. (F)	103	25.1	307	74.9
Q14	In high doses, codeine causes more nausea and vomiting than morphine. (T)	162	39.5	248	60.5
Q15	Suffering and physical pain are synonymous. (F)	122	29.8	288	70.2
Q16	Demerol (Pethidine) is not an effective analgesic in the control of chronic pain. (T)	112	27.3	298	72.7
Q17	The accumulation of losses renders burnout inevitable for those who seek work in palliative care. (F)	84	20.5	326	79.5
Q18	Manifestations of chronic pain are different from those of acute pain. (T)	279	68.0	131	32.0
Q19	The loss of a distant or contentious relationship is easier to resolve than the loss of one that is close or intimate. (F)	80	19.5	330	80.5
Q20	The pain threshold is lowered by anxiety or fatigue. (T)	275	67.1	135	32.9
Total PCQN correct rate 37.1%					

PCQN: palliative Care Quiz for Nurses; T: the answer of the question is “true”; F: the answer of the question is “false”.

**Table 3 ijerph-19-09563-t003:** Nursing students’ attitudes toward the care of the dying patient (n = 410).

Item	Mean *	SD	Weighted%
I would be uncomfortable talking about impending death with the dying person. (R)	2.70	1.076	54%
2. I would not want to care for a dying person. (R)	3.59	1.064	71.8%
3. The non-family caregivers should not be the one to talk about death with the dying person. (R)	3.29	0.961	65.8%
4. I would be upset when the dying person I was caring for gave up hope of getting better. (R)	2.47	1.104	49.4%
5. It is difficult to form a close relationship with the dying person. (R)	3.25	0.982	65.0%
6. When a patient asks, “Am I dying?” I think it is best to change the subject to something cheerful. (R)	2.73	1.154	54.6%
7. I am afraid to become friends with a dying person. (R)	2.96	1.077	59.2%
8. I would feel like running away when the person actually died. (R)	2.75	1.194	55.0%
9. I would be uncomfortable if I entered the room of a terminally ill person and found him or her crying. (R)	2.20	0.997	44.0%
FATCOD-B total score range (9–45)	25.94	4.72	57.6%

FATCOD-B: The Short version of Frommelt Attitudes Towards Care of the Dying; * Mean score out of 5 points. (R): Reverse items.

**Table 4 ijerph-19-09563-t004:** Generalized linear regression model for factors associated with students’ knowledge about palliative care and attitudes toward care of dying.

Variable	β	SE	95% CI	Wald	*p*
Knowledge about PC					
Gender					
Male	−0.35	0.30	−0.94–0.23	1.38	0.239
Female	Ref	-	-	-	-
Year of study					
2nd year	−0.22	0.38	−0.98–0.52	0.352	0.550
3rd year	−0.067	0.39	−0.83–0.70	0.029	0.868
4th year	Ref	-	-	-	-
Pervious lectures & training in PC					
Yes	1.00	0.30	0.41–1.59	11.14	0.001
No	Ref	-	-	-	-
Witnessed dying patient					
Yes	0.58	0.33	−0.11–1.06	2.52	0.081
No	Ref	-	-	-	-
Caring of relative in their last days of life					
Yes	0.47	0.30	−0.11–1.06	2.52	0.112
No	Ref	-	-	-	-
Participation in the preparation of dead patients					
Yes	0.005	0.31	−0.60–0.61	0.000	0.988
No	Ref	-	-	-	-
Age	0.048	0.07	−0.09–0.19	0.42	0.513
GPA	0.03	0.02	−0.00–0.07	2.61	0.016
**Attitudes towards care of dying**					
Gender					
Male	2.07	0.477	1.14–3.01	19.94	<0.001 *
Female	Ref	-	-	-	-
Year of study					
2nd year	1.487	0.62	0.26–2.71	5.65	0.017
3rd year	−0.137	0.63	−1.39–1.11	0.046	0.831
4th year	Ref	-	-	-	-
Previous lectures and training in PC					
Yes	1.410	0.48	0.45–2.36	8.37	0.004
No	Ref	-	-	-	-
Witnessed dying patient					
Yes	0.129	0.54	−0.94–1.20	0.055	0.814
No	Ref	-	-	-	-
Caring of relative in their last days of life					
Yes	0.962	0.488	0.004–1.921	3.87	0.049
No	Ref	-	-	-	-
Participation in the preparation of dead patients					
Yes	0.398	0.50	−0.59–1.38	0.622	0.430
No	Ref	-	-	-	-
Age	0.061	0.119	−0.17–0.29	0.262	0.609
GPA	−0.028	0.32	−0.09–0.035	0.262	0.383

GPA: Grade Point Average; PC: palliative care.

## Data Availability

The data presented in this study are available on request from the corresponding author.

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
