# Peer review of "Palliative Care Knowledge and Attitudes towards End-of-Life Care among Undergraduate Nursing Students at Al-Quds University: Implications for Palestinian Education"

_ijerph, 2022, doi:10.3390/ijerph19159563_

Round 1
Reviewer 1 Report
Thank you for submitting your article for review with the Journal.
Abstract
Line 19 insert "who" between students participated.
Line 20 define FATCOD. With your use of the word "adopted" do you mean "used" if so consider changing.
Line 24 please put the range for the attitude score (as you have done for the knowledge score).
Introduction
Line 42 consider deleting "instead"
Line 56 consider change "with' to "at"
Line 76 "still shyly at" does not make sense
Line 81 you refer to ref 17 as a previous study of PC in Palestine it would be useful to put the date of this study in the text (saves the reader having to refer to the article)
Instruments
Line 133 Please describe the FATCOD B in more detail. It is a modification of the original questionaire. Has the scoring changed? Initially the scoring had positive and negative attitude questions and some "reverse scoring" was required of the Likert scales. Please indicate if this has changed in this short form. I note in the 9 questions there are still positive and negative questions. Does this affect scoring? Please describe the anchors to the Likert scale.
Line 137 the must be a typo PCQN should be FATCOD.
Table 1 Cumulative average what do the percentages mean under the Categories column mean/refer to?
Line 186 "your" ?? should be "our"
Discussion
Line 200 define GPA
Line 245 typo "convenience"
In discussion or in the conclusion you could discuss the link between Knowledge and attitude and whether increased knowledge is likely to improve attitude.
You have shown that in your cohort attitude is reasonably good especially versus knowledge.
Is this all due to the Islamic faith or are there other factors.
If you focussed primarily on attitudes to pain versus knowledge of pain would the result be the same?.
If your University agrees to include PC teaching throughout the course and Knowledge improves will Attitude improve further?.
You will not know the answers to these questions but this type of K&A study is very useful in exploring these questions.
Author Response
25 July. 2022
Manuscript ID: ijerph-1809324, entitled “Undergraduate Palestinian Nursing Students’ Knowledge about Palliative Care and Attitudes towards End-of-Life Care: A Cross-Sectional Study”.
Dear Editor and reviewers,
Thank you for allowing us to revise our manuscript and to respond to the helpful comments raised by the reviewers. Below is a detailed response to comments and changes made to the manuscript.
Response to Reviewer (3)
Thank you for giving me the opportunity to review this interesting article. Overall, the paper is well written with minor English language mistakes.
I have a few comments for the authors to consider
Comment (1): Title: Please consider revising the title as the study only included nursing students from Al Quds university and it is not a representative sample of Palestinian nursing students.
Response: First, we would like to thank you for your fruitful comments that have improved our paper. The title has been modified as “Palliative Care Knowledge and Attitudes towards End-of-Life Care Among Undergraduate Nursing Students at Al-Quds University: Implications for Palestinian Education”
Abstract
Comment (2): Line 17: “their attitudes” please delete “their”
Response: “their” has been deleted.
Comment (3): Line 20: please define the acronym FATCOD
Response: The acronym has been defined.
Introduction
Comment (4): Please revise line 89-91 to highlight that you are identifying gaps and areas for improvement in teaching nursing students at Al Quds university. Your findings cannot be generalized beyond nursing school at Al Quds university.
Response: Lines have been revised to shed light on student al Al-Quds university.
Methods
Comment (5): You do not really need “settings” as a stand alone subheading. You can merge it with procedures.
Response: Settings section has been merged with procedure section.
Comment (6): In section 2.4 (procedure) I highly recommend the authors to provide a brief description of PCQN and FATCOD as they are introduced prior to this section.
Response: A brief description has been added based on your recommendation.
Comment (7): Please provide an interpretation of internal consistency results and how it was calculated in the data analysis section.
Response: An interpretation of internal consistency results has been added in instruments section. Moreover, a paragraph has been added in data analysis section about the interpretation of internal consistency.
Comment (8): Did you administer the questionnaire in English? or did you translate it to local language
Response: English version was used for data collection. Modification has been done in the “instruments” section to highlight that English version was used.
Comment (9): Line 124: “From participants” no need for capital F
Response: Modified.
Results
Comment (10): Line 168-169: please indicate the table number instead of Table X
Response: It has been corrected to “Table 2”.
Comment (11): Table 1: under year of study, please revise the % as they add to 99.9% instead of 100%
Response: The section has been modified.
Comment (12): Please use p value <0.001 instead of 0.000 in text and in table 4
Response: Thank you so much for your comment. p value <0.001 has been used.
Comment (13): In the methods section, the authors mentioned that they have done t test and ANOVA test, but I could not see the results of these analyses in the results section.
Response: An independent sample t-test and ANOVA were used to examine the differences among knowledge, attitude and demographic variables; and the significant findings from these tests were included in linear regression model. So we just present the regression model (table 4) instead on covariate analysis. A sentence has been added to the “data analysis” section to be clearer to the reader as “All variables with a p-value less than 0.25 in univariate analysis were selected for generalised linear regression to identify the influencing variables on knowledge and attitudes. P < 0.05 was considered statistically significant”. Also, a modification has been done in the result section 3.4.
Comment (14): In Table2: I suggest to rename “incorrect answer” column to incorrect answer/did not know.
Response: Thank you for your suggestion, done.
Limitations section
Comment (15): Is it possible that students tried to look up the answers on the internet while filling the questionnaire? how did the authors control this? I think this is worth mentioning the limitations
Response: Agree with your crucial comments, which is one of the limitations of the online survey. A new sentence has been added to the limitation section about the answering the questionnaire online.
Conclusion section
Comment (16): I still believe that authors cannot generalize findings and say that Palestinian nursing students have insufficient knowledge about PC. The sample came from only one nursing school.
Response: The conclusion has been modified focusing on nurses from Al-Quds university.
Additional clarifications
In addition to the above comments, all spelling and gramatical errors pointed out have been corrected. The revised manuscript conforms to the journal style.
We are happy to provide any further clarification if necessary. We hope the above changes have now fully addressed the very useful comments made by the reviewers, and we appreciate their input and time on this, which helped us improve our paper.
Sincerely,
Hammoda Abu-Odah
(on behalf of all authors)

Reviewer 2 Report
The manuscript is relevant and brings contributions to the advancement of nursing science in palliative care.
The authors could have commented in more detail on the inclusion and exclusion criteria of study participants.
Please see attachment.

Author Response
25 July. 2022
Manuscript ID: ijerph-1809324, entitled “Undergraduate Palestinian Nursing Students’ Knowledge about Palliative Care and Attitudes towards End-of-Life Care: A Cross-Sectional Study”.
Dear Editor and reviewers,
Thank you for allowing us to revise our manuscript and to respond to the helpful comments raised by the reviewers. Below is a detailed response to comments and changes made to the manuscript.
Response to Reviewer (2)
The manuscript contains significant information that warrants publication. Although studies have already been published on this topic, it is relevant that undergraduate nursing students are evaluated regarding their knowledge about palliative care. Furthermore, the authors mention that only one similar study was conducted in Palestine.
Comment (1): The title, although a bit long, is consistent with the purpose of the study and the methods used. The summary presents enough information to understand the procedures performed and their outcome. The descriptions are consistent with the manuscript and will help to identify and increase its visibility. The introduction is clear, presents a logical sequence and justifies the importance of the study.
Response: Thank you so much for your positive feedback to our paper.
Comment (2): The method is consistent with the title and objective of the study. The description of the methodological procedures is appropriate to the type of study. The authors used the STROPE framework, from the EQUATOR network. The place and period of the study, and participants involved were adequately described (the authors could have explained the inclusion and exclusion criteria better). Regarding the data collection instruments used, it would be interesting for the authors to comment on whether they were validated for the local culture and language (I believe not, but this must be described in the manuscript, with the justification)
Response: The inclusion and exclusion criteria have been explained better in this version. For instruments, no translation to Arab language has been done for many reasons that presented in the manuscript as “The instruments were distributed in English, as the students teaching courses were in English. Both instruments were previously used in Arab context which were valid to the Palestinian culture. However, the using English version may hinder some students to understand the questionnaire comprehensively which is one of the limitations of the study. A new sentence has been added to the limitation section as “Using the English version of the questionnaire could hinder student nurses to comprehension understanding of some items”.
Comment (3): The results are limited to the findings obtained and the tables presented facilitate the understanding of the data. The data presented in the first and second paragraphs of the results section could have been omitted, considering that such data are already presented in table 1. In item 3.2 of the results, the authors mentioned “Table X”, but it was not presented. (I think it was a mistake by the author and should be corrected.
Response: Thank you so much for your comments in this section.The first paragraph presented the response rate and some statistics, so if you do not mind we would like to keep them. “Table X” has been corrected to “Table 2”
The discussion addresses the relationship between the data obtained in this study and the national and international literature, emphasizing agreements and divergences with other studies that have already been published. The authors present the limitation of the study, but they have better emphasized the contributions of their study of nursing.
Response: A new paragraph has been added to the discussion section, presenting the contribution of study in nursing.
The conclusion responds to the proposed objective and is based on the results and it consistent with the title and method.
Response: Thank you so much.
References are adequate and 65% of them data from the last five years
Response: Thank you so much.
Additional clarifications
In addition to the above comments, all spelling and gramatical errors pointed out have been corrected. The revised manuscript conforms to the journal style.
We are happy to provide any further clarification if necessary. We hope the above changes have now fully addressed the very useful comments made by the reviewers, and we appreciate their input and time on this, which helped us improve our paper.
Sincerely,
Hammoda Abu-Odah
(on behalf of all authors)

Reviewer 3 Report
Thank you for giving me the opportunity to review this interesting article. Overall, the paper is well written with minor English language mistakes.
I have a few comments for the authors to consider
Title: Please consider revising the title as the study only included nursing students from Al Quds university and it is not a representative sample of Palestinian nursing students .
Abstract
Line 17: “their attitudes” please delete “their”
Line 20: please define the acronym FATCOD
Introduction
- Please revise line 89-91 to highlight that you are identifying gaps and areas for improvement in teaching nursing students at Al Quds university. Your findings cannot be generalized beyond nursing school at Al Quds university.
Methods
- You do not really need “settings” as a stand alone subheading. You can merge it with procedures.
- In section 2.4 (procedure) I highly recommend the authors to provide a brief description of PCQN and FATCOD as they are introduced prior to this section.
- Please provide an interpretation of internal consistency results and how it was calculated in the data analysis section.
- Did you administer the questionnaire in English ? or did you translate it to local language
- Line 124: “From participants” no need for capital F
Results
- Line 168-169: please indicate the table number instead of Table X
- Table 1: under year of study, please revise the % as they add to 99.9% instead of 100%
- Please use p value <0.001 instead of 0.000 in text and in table 4
- In the methods section, the authors mentioned that they have done t test and ANOVA test, but I could not see the results of these analyses in the results section.
- In Table2: I suggest to rename “incorrect answer” column to incorrect answer/did not know.
Limitations section
- Is it possible that students tried to look up the answers on the internet while filling the questionnaire? how did the authors control this ? I think this is worth mentioning the limitations
Conclusion section
- I still believe that authors cannot generalize findings and say that Palestinian nursing students have insufficient knowledge about PC. The sample came from only one nursing school.
Author Response
25 July. 2022
Manuscript ID: ijerph-1809324, entitled “Undergraduate Palestinian Nursing Students’ Knowledge about Palliative Care and Attitudes towards End-of-Life Care: A Cross-Sectional Study”.
Dear Editor and reviewers,
Thank you for allowing us to revise our manuscript and to respond to the helpful comments raised by the reviewers. Below is a detailed response to comments and changes made to the manuscript.
Response to Reviewer (1)
Thank you for submitting your article for review with the Journal.
Abstract
Comment (1): Line 19 insert "who" between students participated.
Response: First, we would like to thank you for your fruitful comments that have improved our paper. “Who” has been inserted as your suggestion.
Comment (2): Line 20 define FATCOD.
Response: The FACTOD abbreviation has been defined.
Comment (3): With your use of the word "adopted" do you mean "used" if so consider changing.
Response: The word “Adopted” has been changed to “used”
Comment (4): Line 24 please put the range for the attitude score (as you have done for the knowledge score).
Response: The range of attitude has been added as your request.
Introduction
Comment (5): Line 42 consider deleting "instead"
Response: "instead"has been deleted.
Comment (6): Line 56 consider change "with' to "at"
Response: Change has been done.
Comment (7): Line 76 "still shyly at" does not make sense
Response: The word “shyly” has been changed to “weak”
Comment (8): Line 81 you refer to ref 17 as a previous study of PC in Palestine it would be useful to put the date of this study in the text (saves the reader having to refer to the article)
Response: The year of study “2015” has been added to the text.
Instruments
Comment (9): Line 133 Please describe the FATCOD B in more detail. It is a modification of the original questionaire. Has the scoring changed? Initially the scoring had positive and negative attitude questions and some "reverse scoring" was required of the Likert scales. Please indicate if this has changed in this short form. I note in the 9 questions there are still positive and negative questions. Does this affect scoring? Please describe the anchors to the Likert scale.
Response: Thank you so much for highlighting this point. More description about the scale have been presented in the “Instrument” section. Also, in Table (3), the negative FACTOD items were highlighted.
Comment (10): Line 137 the must be a typo PCQN should be FATCOD.
Response: Agree, sorry for the typo; it has been changed to FATCOD.
Comment (11): Table 1 Cumulative average what do the percentages mean under the Categories column mean/refer to?
Response: Cumulative average refers to the students’ Grade Point Average. It has been modified to be more cleared as “Cumulative grade point average”
Comment (12): Line 186 "your" ?? should be "our"
Response: Its has been modified to “our”. Thank you.
Discussion
Comment (13): Line 200 define GPA
Response: GPA “Grade Point Average” has been defined.
Comment (14): Line 245 typo” convenience"
Response: Modified.
Comment (15): In discussion or in the conclusion you could discuss the link between Knowledge and attitude and whether increased knowledge is likely to improve attitude. You have shown that in your cohort attitude is reasonably good especially versus knowledge. Is this all due to the Islamic faith or are there other factors. If you focussed primarily on attitudes to pain versus knowledge of pain would the result be the same?. If your University agrees to include PC teaching throughout the course and Knowledge improves will Attitude improve further?. You will not know the answers to these questions but this type of K&A study is very useful in exploring these questions.
Response: Thank you so much for this point and K&A questions. Sentences have been added in the conclusion and discussion present the link between knowledge and attitudes. The students favourable attitude could be related to Islamic beliefs. A sentence has been addeded as “The favorable attitude among Palestinian nursing students justified the Islamic religious beliefs that acknowledge the inevitability of death and accept death as a normal process”, but the attitudes can be strengthen by enhancing the students knowledge.
Additional clarifications
In addition to the above comments, all spelling and gramatical errors pointed out have been corrected. The revised manuscript conforms to the journal style.
We are happy to provide any further clarification if necessary. We hope the above changes have now fully addressed the very useful comments made by the reviewers, and we appreciate their input and time on this, which helped us improve our paper.
Sincerely,
Hammoda Abu-Odah
(on behalf of all authors)

Round 2
Reviewer 3 Report
I would like to thank the authors for addressing my comments. I have no further comments